# Global Emissions and Abundances of Chemically and Radiatively Important Trace Gases from the AGAGE Network

Luke M. Western<sup>1</sup>, Matthew Rigby<sup>1</sup>, Jens Mühle<sup>2</sup>, Paul B. Krummel<sup>3</sup>, Chris R. Lunder<sup>4</sup>, Simon O'Doherty<sup>1</sup>, Stefan Reimann<sup>5</sup>, Martin K. Vollmer<sup>5</sup>, Dickon Young<sup>1</sup>, Ben Adam<sup>1</sup>, Paul J. Fraser<sup>3</sup>, Anita L. Ganesan<sup>6</sup>, Christina M. Harth<sup>2</sup>, Ove Hermansen<sup>4</sup>, Jooil Kim<sup>2</sup>, Ray L. Langenfelds<sup>3</sup>, Zoë M. Loh<sup>3</sup>, Blagoj Mitrevski<sup>3</sup>, Joseph R. Pitt<sup>1</sup>, Peter K. Salameh<sup>7</sup>, Roland Schmidt<sup>2</sup>, Kieran Stanley<sup>1</sup>, Ann R. Stavert<sup>3</sup>, Hsiang-Jui Wang<sup>8</sup>, Ray F. Weiss<sup>2</sup>, and Ronald G. Prinn<sup>9</sup> <sup>1</sup>School of Chemistry, University of Bristol, Bristol, UK <sup>2</sup>Scripps Institution of Oceanography, University of California San Diego, La Jolla, CA, USA <sup>3</sup>CSIRO Environment, Aspendale, VIC, Australia <sup>4</sup>NILU, Kjeller, Norway <sup>5</sup>Empa, Laboratory for Air Pollution / Environmental Technology, Dübendorf, Switzerland <sup>6</sup>School of Geographical Sciences, University of Bristol, Bristol, UK

<sup>8</sup>School of Earth and Atmospheric Sciences, Georgia Institute of Technology, Atlanta, GA, USA

<sup>9</sup>Center for Sustainability Science and Strategy, Massachusetts Institute of Technology, Cambridge, MA, USA

Correspondence: Luke M. Western (luke.western@bristol.ac.uk)

Abstract. Measurements from the Advanced Global Atmospheric Gases Experiment (AGAGE) combined with a global 12box model of the atmosphere have long been used to estimate global emissions and surface mean mole fraction trends of atmospheric trace gases. Here, we present annually updated estimates of these global emissions and mole fraction trends for 42 compounds measured by the AGAGE network, including chlorofluorocarbons, hydrochlorofluorocarbons, hydrofluorocarbons,

5 perfluorocarbons, sulfur hexafluoride, nitrogen trifluoride, methane, nitrous oxide, and selected other compounds. The data sets are available at https://doi.org/10.5281/zenodo.15372480. We describe the methodology to derive global mole fraction and emissions trends, which includes the calculation of semihemispheric monthly mean mole fractions, the mechanics of the 12-box model and the inverse method that is used to estimate emissions from the observations and model. Finally, we present examples of the emissions and mole fraction datasets for the 42 compounds.

# 10 1 Introduction

Quantifying the emissions of halogenated and other trace gases is crucial for estimating their environmental impacts, such as ozone layer destruction, and for evaluating the progress of mitigation efforts. The Advanced Global Atmospheric Gases Experiment (AGAGE, Prinn et al., 2000, 2018) network publicly releases measurements of the dry-air mole fractions of 45 atmospheric compounds (Prinn et al., 2022). Measurements made through AGAGE and its predecessors (see Section 2) were

15 initially used to derive atmospheric lifetimes of CFC-11, CFC-12 (Cunnold et al., 1983) and other trace gases, (e.g., Prinn et al., 1983a, 1995, 2005; Rigby et al., 2013; Thompson et al., 2024). However, the predominant use of AGAGE measurements

currently is to estimate global emissions and mole fraction trends over time (Section 4). Here, we present global emissions and derived mole fraction trends for 42 trace gases (Table 1), inferred from AGAGE measurements and a 12-box model of the atmosphere (estimates are not provided for hydrogen, carbon monoxide and trichloroethene). We refer to these quantities as AGAGE-derived products, which are available at https://doi.org/10.5281/zenodo.15372480 (Western et al., 2025). The primary purpose of this article is to describe the methodology underpinning these AGAGE-derived products.

Global emissions and mole fraction trends are derived using AGAGE measurements coupled with a two-dimensional 12box model of the atmosphere (Cunnold et al., 1983, 1994; Rigby et al., 2013). This is in contrast to inventory methods, or bottom-up methods, using activity data and emission factors, which can quantify expected emissions. The estimated emissions

presented here, also known as top-down emissions, are inferred from measured mole fraction. The modelled global and semihemispheric mole fractions presented (see Section 5) are also inferred using measurements. The reason for inferring the mole fractions, rather than directly using the measurements themselves, is primarily so that mole fractions can be inferred during times where no measurements are available, e.g., due to instrument downtime.

- AGAGE-derived products of ozone-depleting substances (ODSs) and greenhouse gases (GHGs) have been used in the Scientific Assessments of Ozone Depletion of the World Meteorological Organisation (WMO) (e.g., Ehhalt and Fraser, 1988; Laube and Tegtmeier, 2023; Liang and Rigby, 2023; Daniel and Reimann, 2023) and in the Assessment Reports of the Intergovernmental Panel on Climate Change (e.g., IPCC et al., 1990; Gulev and Thorne, 2023). Emission estimates using AGAGE measurements have been published in many research articles. Some recent notable outputs are the identification of excess CFC-11 emissions in eastern China after its global production phaseout (Montzka et al., 2021; Rigby et al., 2019; Park et al.,
- 2021), of discrepancies in reported abatement and estimated global and Chinese HFC-23 emissions (Stanley et al., 2020; Adam et al., 2024), of a rapid increase in unregulated global and Chinese chloroform emissions (Fang et al., 2019), and of increases in ODS emissions used as feedstock after their phase-out for dispersive uses (Lickley et al., 2021; Vollmer et al., 2018; Western et al., 2023). Emission estimates from the AGAGE network have exposed various unusual or rapidly increasing trends in ODSs (e.g., Vollmer et al., 2015; Liang et al., 2016; Vollmer et al., 2016; Simmonds et al., 2017; Vollmer et al., 2018; An et al
- 2021; Western et al., 2022; An et al., 2023) and halogenated GHGs (Mühle et al., 2009; Miller et al., 2010; Mühle et al., 2010; Rigby et al., 2010; Vollmer et al., 2011; Arnold et al., 2014; Lunt et al., 2015; O'Doherty et al., 2014; Simmonds et al., 2016; Fortems-Cheiney et al., 2015; Simmonds et al., 2018, 2020; Mühle et al., 2022). AGAGE measurements have also been used to identify several compounds in the atmosphere for the first time and to quantify their associated emissions (e.g., ?Mühle et al., 2009; Schoenenberger et al., 2015; Vollmer et al., 2015a, b, 2019, 2021).
- We describe the AGAGE measurements used to derive the derived products in Section 2, the AGAGE 12-box model in Section 3, and the inverse framework in Section 4. A brief description of the contents of the AGAGE derived products is provided in Section 5. The atmospheric budgets derived from the products are then presented in Section 6. Finally, limitations are outlined in Section 7 and a brief summary is given in Section 8.

| Common Name          | Chemical Formula                | Common Name             | Chemical Formula                  |
|----------------------|---------------------------------|-------------------------|-----------------------------------|
| PFC-14               | $CF_4$                          | HCFC-124                | CHClFCF <sub>3</sub>              |
| PFC-116              | $C_2F_6$                        | HCFC-132b               | $CH_2ClCClF_2$                    |
| PFC-218              | $C_3F_8$                        | HCFC-133a               | CH <sub>2</sub> ClCF <sub>3</sub> |
| PFC-318              | c-C <sub>4</sub> F <sub>8</sub> | CFC-11                  | $CCl_3F$                          |
| Sulfur Hexafluoride  | $SF_6$                          | CFC-12                  | $CCl_2F_2$                        |
| Sulfuryl Fluoride    | $SO_2F_2$                       | CFC-13                  | CClF <sub>3</sub>                 |
| Nitrogen Trifluoride | $NF_3$                          | CFC-113/a <sup>1</sup>  | $C_2 C l_3 F_3$                   |
| HFC-23               | $CHF_3$                         | CFC-114/a <sup>2</sup>  | $C_2 C l_2 F_4$                   |
| HFC-32               | $CH_2F_2$                       | CFC-115                 | $CClF_2CF_3$                      |
| HFC-134a             | $CH_2FCF_3$                     | Halon-1211              | $CBrClF_2$                        |
| HFC-152a             | $CH_3CHF_2$                     | Halon-1301              | CBrF <sub>3</sub>                 |
| HFC-125              | $CHF_2CF_3$                     | Halon-2402              | $CBrF_2CBrF_2$                    |
| HFC-143a             | $CH_3CF_3$                      | Methyl Chloride         | $CH_3Cl$                          |
| HFC-227ea            | $CF_3CHFCF_3$                   | Methyl Bromide          | $CH_3Br$                          |
| HFC-236fa            | $CF_3CH_2CF_3$                  | Dichloromethane         | $CH_2Cl_2$                        |
| HFC-245fa            | $CHF_2CH_2CF_3$                 | Chloroform              | $CHCl_3$                          |
| HFC-365mfc           | $CH_3CF_2CH_2CF_3$              | Carbon Tetrachloride    | $\mathrm{CCl}_4$                  |
| HFC-43-10mee         | $CF_3(CHF)_2CF_2CF_3$           | Methyl Chloroform       | CH <sub>3</sub> CCl <sub>3</sub>  |
| HCFC-22              | $CHClF_2$                       | Perchloroethylene (PCE) | $CCl_2=CCl_2$                     |
| HCFC-141b            | $CH_3CCl_2F$                    | Methane                 | $CH_4$                            |
| HCFC-142b            | $CH_3CClF_2$                    | Nitrous Oxide           | $N_2O$                            |

Table 1. Compounds measured by the AGAGE network for which emissions and atmospheric mole fraction trends are estimated

<sup>1</sup> CFC-113/a is a composite of the isomers CFC-113 (CClF<sub>2</sub>CCl<sub>2</sub>F) and CFC-113a (CCl<sub>3</sub>CF<sub>3</sub>). However, the contribution of each isomer to the total mole fraction is not yet well understood.

<sup>2</sup> CFC-114/a is a composite of the isomers CFC-114 ( $CCIF_2CCIF_2$ ) and CFC-114a ( $CCl_2FCF_3$ ). As footnote<sup>1</sup>.

# 2 Measurements

- The AGAGE network and its two predecessors have measured the atmospheric abundance of trace gases since 1978 (Atmospheric Lifetime Experiment, ALE: 1978–1981; Global Atmospheric Gases Experiment, GAGE: 1982–1992; AGAGE: since 1993). AGAGE measurements are combined with the 12-box model and an inverse method to produce the derived products presented here. A complete description of the measurements made by the AGAGE network and its two predecessors is given by Prinn et al. (1983b), Prinn et al. (2000) and Prinn et al. (2018), including detailed descriptions of the measurement instruments
- and the calibration scales used for each trace gas. Here, we only summarise those measurements used as inputs to the 12-box model (Section 3), which are only from a subset of measurement sites, compounds and measurements made in the current

Figure 1. Locations of the AGAGE stations, from which measurements of dry-air mole fraction are currently used to derive the data sets using the 12-box model and an inverse method. Thin grey lines show the equator and at  $\pm 30^{\circ}$ N.

network and its predecessors, with a focus on sites that frequently measure well-mixed background air throughout the year, that is, excluding sites that regularly measure highly polluted air masses.

The data sets described in Section 5 use measurements from five historic AGAGE stations and two newer AGAGE sites:

- Zeppelin (ZEP), Svalbard, Norway (78.9°N, 11.9°E; began 2001), Mace Head (MHD), Ireland (53.3°N, 9.9°W; began 1987), Jungfraujoch (JFJ), Switzerland (46.5°N, 8.0°E; began 2000), Trinidad Head (THD), California, USA (41.1°N, 124.2°W; began continuously in 1995), Ragged Point (RPB), Barbados (13.2°N, 59.4°W; began 1978), Cape Matatula (SMO), American Samoa (14.2°S, 170.6°W; began 1978), and at Kennaook/Cape Grim (CGO), Tasmania, Australia (40.7°S, 144.7°E; began continuously in 1978). Figure 1 shows the location of these stations. The AGAGE network contains many more sites than
- the ones used here. The measurement sites that have been used have been selected due to their usefulness in measuring air that is less impacted by nearby pollution, and therefore more representative of background conditions, and their longevity of measurements.

The data sets presented here are primarily derived from in situ high-frequency measurements (Section 2.1). For a subset of substances, the in situ measurements are complemented by measurements of archived air samples (Section 2.2). Measurements
from the ALE network from two sites – Adrigole (ADR), Ireland (52°N, 10°W) and Cape Meares (CMO), Oregon (45°N, 124°W°W) – are used for CFC-11, CFC-12, CFC-113/a, CCl<sub>4</sub> and CH<sub>3</sub>CCl<sub>3</sub> before measurements from MHD and THD are available. See Prinn et al. (1983b) for more information. Some publications have also used measurements of firn air, collected in Greenland and Antarctica, to derive emissions with the 12-box model (e.g., Trudinger et al., 2016; Vollmer et al., 2016, 2018), but the routinely published data sets presented here currently do not contain measurements made from firn air.

# 75 2.1 High-frequency measurements

Here, we describe the measurements used to derive global emissions and mole fraction trends. AGAGE in situ measurements of ODSs and GHGs have historically been made using multiple measurement instruments at each site. Measurements from Medusa gas chromatography mass spectrometry (GC-MS) systems (Miller et al., 2008; Arnold et al., 2012), deployed at each AGAGE site in the early to late 2000s and 2010s, are used to derive global emissions and mole fraction trends for most of

- the compounds listed in Table 1. Exceptions are CFC-11, CFC-12, CCl<sub>4</sub>, and N<sub>2</sub>O, for which measurements by the AGAGE gas chromatography 'multidetector' (GC-MD) systems at MHD, THD, RPB, SMO and CGO are preferentially used (in the case of N<sub>2</sub>O exclusively) due to higher measurement frequency and longer measurement records (see Prinn et al., 2000). These compounds are measured using electron capture detection (ECD) (Prinn et al., 2000). Sites that do not have a GC-MD instrument (JFJ and ZEP) were not used to estimate CFC-11, CCl<sub>4</sub>, and N<sub>2</sub>O mole fractions and emissions.
- Prior to Medusa GC-MS measurements, some compounds had been measured on GC-MS adsorption-desorption systems (ADS), starting out with a prototype-ADS system at MHD in mid-1994, followed by ADS systems at MHD and CGO in late 1997 (see Simmonds et al., 1995; Prinn et al., 2000, for more information). GC-MS-ADS measurements also commenced at JFJ in 2000 (Reimann et al., 2004, 2008) and at ZEP in 2001 (Platt et al., 2022). Information about the compounds and time periods for which these GC-MS-ADS data are used to derive trends in emissions and mole fractions is contained in Prinn et al.
- (2025). All GC-MD, ADS, and Medusa measurements are reported on the calibration scales used by AGAGE, as detailed in Prinn et al. (2025).

Methane has been historically measured by AGAGE GC-MD systems (and its GAGE predecessor) using flame-ionization detection (FID), reported on the Tohuko 1987 scale maintained at SIO, but at some sites in recent years GC-MD  $CH_4$  measurements have been replaced/superseded by cavity ring-down spectrometer (CRDS) instruments (Picarro) (see Prinn et al.,

2018), reported on the NOAA-2004A scale (Dlugokencky et al., 2005). Extensive NOAA-AGAGE intercomparisons as well as AGAGE on-site instrument comparisons during instrument overlap have shown that the scale differences are negligible (NOAA/AGAGE ratio of 1.0001  $\pm$  0.0007, Prinn et al., 2018). Sites that do not have a GC-MD instrument (JFJ and ZEP) were not used for methane (even when CRDS measurements were available).

The typical repeatability of the measurements made by the GC-MS Medusa and GC-MD systems discussed here range from 0.05% for N<sub>2</sub>O, 0.1% for CF<sub>4</sub> and CFC-12, 0.3-1% for most compounds, and up to 7% for HFC-236fa of the measured standard value. For CH<sub>4</sub>, GC-MDs achieve 0.2% and CRDS systems achieve 0.02% (for more information see Prinn et al., 2018). The

measurement repeatability is mainly compound and detector-dependent and is largely dominated by the atmospheric abundance of the compound but can also be negatively affected by site specific problems such as lab air contamination, lab temperature problems, trap temperature fluctuations, or MS filament problems.

The measurement data sets available from Prinn et al. (2025) provide details on the instruments used to measure each compound listed in Table 1, and for which period. The measurements are used in conjunction with the statistical AGAGE pollution algorithm to determine pollution free monthly mean baseline mole fractions, which are then used as input for the model and inversion to produce the data sets presented here. This method allows for the determination of monthly mean baseline mole fractions, as detailed in Section 2.3.

# 110 2.2 Archived air measurements

For several compounds, measurements of archived air samples are used to extend the in situ measurement record back into the past. For the southern hemisphere, air collection and archiving began in 1978 with the Cape Grim Air Archive (CGAA), where air samples were taken at CGO during clean air conditions with cryogenic methods and stored in stainless steel tanks (Fraser et al., 1991; Langenfelds et al., 1996), with the intention of reconstructing the historical composition of ambient air

- once suitable analytical instruments and calibration scales were developed. Early measurements of the CGAA were performed on various instruments (summarized in Fraser et al., 2018), but here we focus on Medusa GC-MS measurement made in 2007 (e.g., Miller et al., 2010) and 2011 (e.g., Ivy et al., 2012), which were used in many subsequent studies (e.g., Mühle et al., 2009; O'Doherty et al., 2009; Rigby et al., 2010; O'Doherty et al., 2014). Later CGAA measurement made in 2016 (Vollmer et al., 2016) are currently not used here. These Medusa GC-MS measurements were mostly performed at the CSIRO Aspendale
- laboratory in Australia, but also at the Scripps Institution of Oceanography (SIO), in La Jolla, California USA. The frequency of available CGAA air samples differs, with one or two samples per year typically available before 1994, and up to nine samples available per year between 1994-1999, after which measurements of ongoing archived air samples are no longer used in this work. There is good agreement between the measurements at SIO and CSIRO of identical air samples and air samples with the same or similar fill dates.
- To complement the CGAA, archived air samples from the Northern Hemisphere were gathered from several laboratories and mostly measured on Medusa GC-MS systems at SIO. Many of these tanks had been filled at THD or SIO, some at other northern hemispheric locations in the USA (such as Cape Meares in Oregon, Point Barrow in Alaska, and Niwot Ridge in Colorado) between 1973 and 2016 (Mühle et al., 2010, 2009). Unlike the CGAA, many of these samples were not originally intended for future atmospheric archive measurements and required more stringent quality control. For inert and/or volatile
- or very abundant compounds (such as CF<sub>4</sub>, SF<sub>6</sub>, NF<sub>3</sub>, many HFCs and HCFCs) the resulting measurements were well-suited to reconstruct historic northern hemispheric abundances. Measurements of other compounds (e.g., several minor CFCs, H-2402, HCFC-22, HCFC-124, HFC-43-10mee, PFC-218) produced some anomalous data points during data processing, which resulted in less certain northern hemispheric historic abundances for these compounds. Some archived air samples from the northern hemisphere were also measured at CSIRO (Arnold et al., 2012; Ivy et al., 2012; Mühle et al., 2010, 2009), again
- generally confirming that measurements from the instruments at SIO and CSIRO can be combined.

Table 2 shows the compounds that use archived air measurements in their emissions estimates. The references in Table 2 are the first publications in which archived air was used to derive emissions, and subsequent relevant publications.

| Table 2. | Relevant publications of compounds that have used archived air measuremen | ts, alongside high frequency | measurements made by |
|----------|---------------------------------------------------------------------------|------------------------------|----------------------|
| AGAGE    | to quantify emissions estimates.                                          |                              |                      |

| Compounds                       | Reference                                                                                    |
|---------------------------------|----------------------------------------------------------------------------------------------|
| $CF_4$                          | Mühle et al. (2010); Trudinger et al. (2016)                                                 |
| $C_2F_6$                        | Mühle et al. (2010); Trudinger et al. (2016)                                                 |
| $C_3F_8$                        | Mühle et al. (2010); Trudinger et al. (2016)                                                 |
| c-C <sub>4</sub> F <sub>8</sub> | Mühle et al. (2019)                                                                          |
| $SF_6$                          | Rigby et al. (2010); Simmonds et al. (2020)                                                  |
| $NF_3$                          | Arnold et al. (2013)                                                                         |
| $\mathrm{SO}_2\mathrm{F}_2$     | Mühle et al. (2009)                                                                          |
| HFC-23                          | Miller et al. (2010); Simmonds et al. (2018); Stanley et al. (2020)                          |
| HFC-32                          | O'Doherty et al. (2014)                                                                      |
| HFC-125                         | O'Doherty et al. (2009)                                                                      |
| HFC-134a                        | O'Doherty et al. (2004); Rigby et al. (2014)                                                 |
| HFC-143a                        | O'Doherty et al. (2014)                                                                      |
| HFC-152a                        | Simmonds et al. (2016)                                                                       |
| HFC-227ea                       | Vollmer et al. (2011)                                                                        |
| HFC-236fa                       | Vollmer et al. (2011)                                                                        |
| HFC-245fa                       | Vollmer et al. (2011)                                                                        |
| HFC-365mfc                      | Vollmer et al. (2011)                                                                        |
| HFC-43-10mee                    | Arnold et al. (2014)                                                                         |
| HCFC-22                         | O'Doherty et al. (2004); Saikawa et al. (2012); Western et al. (2024b)                       |
| HCFC-141b                       | O'Doherty et al. (2004); Simmonds et al. (2017); Western et al. (2022)                       |
| HCFC-142b                       | O'Doherty et al. (2004); Simmonds et al. (2017); Rigby et al. (2014); Western et al. (2024b) |
| CFC-13                          | Vollmer et al. (2018)                                                                        |
| CFC-115                         | Vollmer et al. (2018)                                                                        |
| CFC-113/a                       | Rigby et al. (2014)                                                                          |
| CFC-114/a                       | Vollmer et al. (2018)                                                                        |
| H-1211                          | Vollmer et al. (2016)                                                                        |
| H-1301                          | Vollmer et al. (2016)                                                                        |
| H-2402                          | Vollmer et al. (2016)                                                                        |

# 2.3 Derivation of baseline mole fractions

Derived global emissions and mole fraction trends are inferred from monthly mean 'baseline' mole fractions for each measurement site in Section 2. A baseline measurement is when the sampled air is well mixed within the air parcel and is not influenced by nearby pollution sources. These monthly mean baseline mole fractions are fed into the inversion framework described in Section 4.

Here, monthly mean baseline measurements are derived using a statistical algorithm (O'Doherty et al., 2001). The algorithm identifies measurements that are considered as baseline by taking the following steps.

- 1. For a given day, fit a second-order polynomial to the daily minima of measurements over a 121-day window centred on that day (i.e. using 60 days before and after). Subtract the fitted polynomial from all measurements within the window, to detrend the measurements, and calculate the median of these detrended data. Calculate the root mean square error (RMSE) using only the detrended values that fall below this median value. Classify measurements on the given day as baseline if they are within three times the RMSE of the median. Compute this step as a moving window across all days.
- 2. Repeat step 1 using the resultant tentative baseline measurements, with initial pollution events removed, from the first iteration of Step 1. Additionally, during this step, label measurements that fall within two to three times the new RMSE as 'possibly polluted' measurements.
  - 3. Remove the 'possibly polluted' measurements if the following or preceding measurement is also labelled as a polluted measurement (i.e., greater than three times the RMSE) following Step 2.
- 4. The mean of the remaining baseline measurements for each calendar month is taken as the monthly mean baseline for a given measurement site.

### **3** AGAGE 12-box model

The AGAGE 12-box model is a two-dimensional model that simulates transport of long-lived trace species in the zonal mean atmosphere (i.e., with no longitudinal component). Each trace gas is assumed to be uniformly mixed within each box. The current AGAGE 12-box model has evolved from the 9-box model originally described in Cunnold et al. (1983), which was later expanded to 12 boxes (Cunnold et al., 1994). Several subsequent publications have recoded this original model (Rigby et al., 2013), updated transport parameters or losses (e.g., Rigby et al., 2008), or developed a model adjoint (Thompson et al., 2018). The 12-box model is divided into latitudinal semi-hemispheres at the equator and 30 °N and 30 °S, and vertically at 500 and 200 hPa (with the surface at 1000 hPa), approximating boxes bounded at the planetary boundary layer and tropopause. See

Figure 2 for a schematic representation. The air masses of the four boxes are equal at each vertical level. The 12-box model is governed by source, transport and loss processes, which are described in the remainder of this section.