# Peer review of "Global Emissions and Abundances of Chemically and Radiatively Important Trace Gases from the AGAGE Network"

_Earth System Science Data, 2025_

## Author Comment (AC1)

We thank the reviewer for their comments on the manuscript. Below we address the concerns and comments.

**Why didn't the authors use MALTA, recently developed by Western et al. (2024), for the emission estimations in this paper? What are the differences in emission estimates for each compound between MALTA and the 12-box model? Does this mean that MALTA will not be used for this type of global emission estimation?**

This manuscript describes the data sets produced using a 12-box model and AGAGE measurements. These data sets have a long history of use, which we anticipate will continue. We state in the manuscript that *"More complex transport models combined with AGAGE measurements are likely to complement the data sets provided here (e.g., Western et al., 2024, Liu et al., 2024), although we anticipate that the 12-box model will remain in use for many years to come, due to its efficiency and ease of use in this application.",* which acknowledges that the MALTA model will likely be used in future to estimate emissions. At current, emissions estimates using MALTA cannot be routinely produced in a similar manner to those using the 12-box model.

**Do researchers still need to contact the principal investigators (PIs) of the measurement stations if they cite this paper to use the global mole fractions or emission estimates? Please clarify in the paper.**

The data sets are distributed under a CC BY 4.0 licence, which 'allows re-distribution and re-use of a licensed work on the condition that the creator is appropriately credited.' The data sets are published subject to the AGAGE data policy, which states that station PIs should be contacted to 'ensure that the quality and limitations of the data are accurately represented.' We have added the relevant information to the Code and Data Availability Section: *"All AGAGE derived data sets presented in the paper are available at https://doi.org/10.5281/zenodo.15372480 (Western et al., 2025). Users must agree to the AGAGE Data Policy, the details of which can be found when downloading the data sets."* The AGAGE Data Policy is currently under review, and we therefore would prefer that users refer to the most up to date policy rather than explicitly and persistently quoting it in the manuscript.

**Each AGAGE baseline station is not located at the geometric center of its corresponding surface box in the 12-box model. This spatial mismatch could lead to representational errors, especially for species with regional emission gradients or temporal variability. How do the authors account for this spatial mismatch? And how does it affect the derived global mean mole fractions and the resulting emission estimates?**

We acknowledge this shortcoming of the modelling approach, which we state in Section 7, Limitations. Here, we expand this statement to also include within-box variability, and we now state that "*our derived emissions estimates are sensitive to potential biases in the observations and model. Estimates are available for the uncertainty due to the assumed atmospheric lifetime and calibration scale, and these terms are included in our derived emissions estimates However, for some compounds, particularly those with shorter lifetimes, unaccounted-for biases may exist because the network and model cannot resolve zonal gradients or meridional gradients within each box. For example, a difference between AGAGE and NOAA-derived dichloromethane emissions is thought to be partly due to differences in measurement locations in the Northern Hemisphere tropics between the two networks, as well as a large (~10%) difference in calibration scales (Carpenter and Reimann, 2014).*

**The document mentions the use of various instruments such as GC-MD, ADS, and Medusa GC-MS during different periods. However, the description of data continuity validation during instrument transitions (e.g., from ADS to Medusa) is insufficient. It is recommended to supplement the results of comparisons between different instruments during overlapping observation periods, quantify the range of deviations, and especially clarify the long-term consistency of methane observations between the GC-MD and CRDS systems (beyond the mentioned ratio of 1.0001±0007). Additionally, the impact of calibration scale conversion on historical data should be addressed.**

We thank the reviewer for their attention to the robustness of the continued measurements. When writing the manuscript, the authors made a conscious decision to focus the manuscript on the derived data products, rather than the data quality of the underlying mole fraction measurements, including the rationale behind the now published recommended instrument combinations. The reason for this is twofold: 1) a focus on the input measurements would require a lengthy addition to an already long manuscript and detract from the dataset that is being presented and 2) that an update to Prinn et al., (2018) is currently in preparation, which will provide the rationale and validation for the recommended input measurement data sets, including calibration and instrumentation differences. Please note that the uncertainty associated with errors in calibration scales are propagated into the derived datasets, with the assumed errors detailed in Table S1.

**Section 7.1 mentions that "interannual repetition of meteorological data could lead to emission interannual variability errors," but the impact of this issue has not been assessed. It is suggested to add a sensitivity test (e.g., comparing results using interannual varying meteorological fields) or cite Rigby et al. (2008) to quantify the uncertainty range.**

Deriving annually varying transport fields is beyond the scope of the current manuscript. Uncertainties from annually repeating meteorological fields are propagated into our final estimates as a systematic error, which we state in Line XX, *"The systematic component of transport error is assumed to be 1% of emissions for all substances (one standard deviation)"* .

In Section 7.1, we cite Rigby et al., (2008), and other relevant literature, as suggested, *"Due to the lack of interannually varying OH and other sinks, longer term trends in emissions and year-to-year differences may be misrepresented (e.g., Rigby et al., 2008, 2017; Turner et al., 2017; Naus et al., 2019)."*

**The transport parameters and loss processes (such as OH reactions and stratospheric losses) in the 12-box model are based on the latest literature (Burkholder and Hodnebrog, 2023), and the parameterization scheme is reasonable. The attention given to the prior emission independence in the Bayesian inversion framework (to avoid self-cycling of AGAGE data) is commendable. However, the prior assumption for some compounds (e.g., CFC-13, assumed to be 1/7 of CFC-115) lacks direct validation. It is recommended to include a sensitivity analysis to address this.**

It is unclear what sensitivity study is proposed. At present, the current estimate for CFC-13 emissions remains the most informed estimate we have. The only possible validation that we can conceive comes from the measurement-derived emissions, which would create circularity if we adjust the a priori emissions following our estimate. We have expanded the discussion in the text to *"To the best of our knowledge, no comprehensive inventory of global emissions of CFC-13 exists, so we assume that the a priori emissions for CFC-13 are a seventh of those of CFC-115 (see the supplementary information of Vollmer et al. (2018) for the rationale behind this approximation, which is based on available ratios of production of the two compounds)"*

**The term "Semihemispheric" is inconsistently spelled (Section 5 vs. Section 4.1 "Semi-hemispheric"). The manuscript should standardize this term as "semihemispheric" following AGAGE conventions.**

This has been changes accordingly throughout.

**Saunois et al. (2025) has been officially published (ESSD, 2025); citation information should be updated (volume, issue, pages).**

Thank you, this has been changed.

**Line 43-44: For the sentence "g., ?Mühle et al., 2009", it seems like there is an issue with the citation, as the question mark (?) is likely a placeholder or an error. It should be checked and corrected.**

Thank you. This should be a reference to Weiss et al. (2008).

**Line 70-71: The original sentence "Oregon (45°N, 124°W°W)" contains an error with the extra "°W".**

Thank you. This has been fixed.

**The hyphen used to connect years is inconsistent throughout the text; some instances use "–"while others use "-". For example, in Line 195 and Line 199, it should be unified to "–" throughout the entire manuscript.**

This has been fixed throughout.

---

## Author Comment (AC2)

We thank the reviewer for their comments on the manuscript. Below we address the concerns and comments.

**1. Introduction section. While these listed gases (in Table 1) might be obvious to the authors and folks in the community why they are important and why measured, this isn't necessary clear to the general readers outside the ODS community.  To many, it reads like a long list of chemical compounds, mostly synthetic. It would be very helpful if you can add a paragraph describe these gases in groups, if possible, add the connection between groups.  For example, major greenhouse gases, CFCs, HCFCs, HFCs, PFCs, and HFCs are ozone-friendly alternatives to CFCs and HCFCs, etc.  This way, it helps the readers to understand why they are the target gases measured by AGAGE.**

 We have altered the first paragraph of the introduction so that it is much more descriptive of these trace gases. The first paragraph is now:

*Quantifying the global emissions of halogenated and other long-lived radiatively and chemically important trace gases is crucial for estimating their environmental impacts, such as depletion of the stratospheric ozone layer, and contributions to radiative forcing. Chlorofluorocarbons (CFCs), halons, and the solvents carbon tetrachloride (CCl4) and methyl chloroform (CH3CCl3) are trace gases that have been phased out for emissive use under the Montreal Protocol on Substances that Deplete the Ozone Layer. Emissions of these gases persist because they are still contained within appliances, foams, and other applications, produced before their phase out, and continue to leak into the atmosphere. In some cases, production is ongoing because of their exempted production for chemical manufacture. Some of these substances remain in the atmosphere for years to centuries after they are emitted, owing to their long atmospheric lifetimes. Where non-ozone-depleting alternatives could not immediately be found, these gases were replaced by hydrochlorofluorocarbons (HCFCs), which are currently being phased out under the Montreal Protocol. HCFCs are in turn being replaced by hydrofluorocarbons (HFCs). While HFC do not deplete ozone, they have large global warming potentials, much like the ozone-depleting substances that they replaced. As a result, the production of HFCs is now being phased down under the Kigali Amendment to the Montreal Protocol. Chlorinated very short-lived substances (Cl-VSLS), with atmospheric lifetimes less than around six months, and some halomethanes, with both natural and anthropogenic sources, are not controlled under the Montreal Protocol and may present a threat to ozone layer recovery. Collectively, the controlled and uncontrolled ozone-depleting substances are responsible for almost all of the anthropogenic chlorine and bromine input to the stratosphere. There are a number of non-ozone depleting fluorocarbons that have extremely large global warming potentials, such as perfluorocarbons (PFCs), sulfur hexafluoride (SF6) and nitrogen trifluoride (NF3), and are almost entirely industrially produced. Halogenated substances, along with methane (CH4),*

*which critically affects the oxidative capacity of the atmosphere, and nitrous oxide (N2O), which is also an ozone-depleting substance, are responsible for almost all the gaseous radiative forcing from anthropogenic sources beyond that of carbon dioxide. As a result, monitoring of these gases is crucial to understand the state of the atmosphere.*

**2. Table 4. I would suggest the authors consider include an additional column of the tropospheric OH lifetime for all listed gases that derived with the 12-box model, in addition to the OHa and OHe/r values. This would be useful information in addition to the derived global emissions and can be compared with results from other published literature.**

 Good suggestion, thanks. These have been added to Table 4.

**3. Section 4.2 A priori emissions. It is very nice to see the authors spend a tremendous amount of effort in gathering all possible bottom-up emissions and assemble into an a priori emissions inventory. I noticed that in the data availability section, only AGAGE-derived data sets are made available. I would strongly recommend that authors consider adding the a priori emissions into the data record as well, if possible.**

These will be released alongside the derived products in the next release.

**Isn't it more accurate to say "have been measuring" instead of "have measured"?**

This has been changed.

**L52-53. "AGAGE measurements are combined with the 12-box model and an inverse method to produce the derived products presented here". This sentence is out of place. It should be moved to a later section. May be section 3?**

This has been moved to section 3.

**"have been used have been selected"? you only need one, either used or selected.**

This have been changed to "have been selected".

**Is it possible to include in the supplementary a list of other sites (that are not included in this paper) and the related details?  Some readers might find those information valuable, if in the future this becomes the AGAGE-goto paper?**

This manuscript intends to be a description of derived products from the AGAGE network using a 12-box model. We intentionally do not want this to become the

'AGAGE-goto paper'. There is currently a manuscript in preparation, as an update to Prinn et al. (2018), which will include the requested details. We would prefer to save this information for that manuscript.

**L112, L125 & L134. You should use southern and northern hemisphere consistently, either all start with upper cases or all with lower cases.**

Thank you. This has been changed to upper case, e.g., Northern Hemisphere.

**L235 & L242. Consider change "a priori set of emission estimates" to "a priori emissions". It is a bit redundant as you also use an estimate of emissions right after this.**

Thanks, this has been changed as suggested.

**L243-L244. I don't think you need this sentence. It is a bit hand-waving and unnecessary. Just end with "available bottom-up estimates" in the previous sentence is adequate.**

OK, deleted.

---

## Author Response (AR2)

Dear Yuqiang Zhang,

Please find uploaded the final manuscript.

The colour schemes have been changes in Figures 5 and 7 to make them colour blind friendly.

I have also added a single line to the abstract, stating that the data sets are quantified through 2023, as I realized that this important detail was missing.

Yours,

Luke Western